# The Combined Effects of Television Viewing and Physical Activity on Cardiometabolic Risk Factors: The Kardiovize Study

**DOI:** 10.3390/jcm11030545

**Published:** 2022-01-22

**Authors:** Geraldo A. Maranhao Neto, Iuliia Pavlovska, Anna Polcrova, Jeffrey I. Mechanick, Maria M. Infante-Garcia, Jose Medina-Inojosa, Ramfis Nieto-Martinez, Francisco Lopez-Jimenez, Juan P. Gonzalez-Rivas

**Affiliations:** 1International Clinical Research Center (ICRC), St Anne’s University Hospital (FNUSA) Brno, 656 91 Brno, Czech Republic; iuliia.pavlovska@fnusa.cz (I.P.); anna.polcrova@fnusa.cz (A.P.); maria.garcia@fnusa.cz (M.M.I.-G.); juan.gonzalez@fnusa.cz (J.P.G.-R.); 2Department of Public Health, Faculty of Medicine, Masaryk University, 601 77 Brno, Czech Republic; 3Research Centre for Toxic Compounds in the Environment (RECETOX), Masaryk University, 601 77 Brno, Czech Republic; 4The Marie-Josée and Henry R. Kravis Center for Cardiovascular Health at Mount Sinai Heart, Icahn School of Medicine at Mount Sinai, New York, NY 10029, USA; jeffreymechanick@gmail.com; 5Foundation for Clinic, Public Health, and Epidemiology Research of Venezuela (FISPEVEN INC), Caracas 3001, Venezuela; nietoramfis@gmail.com; 6Division of Preventive Cardiology, Department of Cardiovascular Medicine, Mayo Clinic, Rochester, MN 55905, USA; MedinaInojosa.Jose@mayo.edu (J.M.-I.); lopez@mayo.edu (F.L.-J.); 7Department of Global Health and Population, Harvard TH Chan School of Public Health, Harvard University, Boston, MA 02138, USA; 8LifeDoc Health, Memphis, TN 38119, USA

**Keywords:** television viewing, sedentary behavior, physical activity, cardiometabolic risk factors

## Abstract

The aim of the present study was to evaluate the association between television viewing/physical activity (TVV/PA) interactions and cardiometabolic risk in an adult European population. A total of 2155 subjects (25–64 years) (45.2% males), a random population-based sample were evaluated in Brno, Czechia. TVV was classified as low (<2 h/day), moderate (2–4), and high (≥4). PA was classified as insufficient, moderate, and high. To assess the independent association of TVV/PA categories with cardiometabolic variables, multiple linear regression was used. After adjustments, significant associations were: High TVV/insufficient PA with body mass index (BMI) (β = 2.61, SE = 0.63), waist circumference (WC) (β = 7.52, SE = 1.58), body fat percent (%BF) (β = 6.24, SE = 1.02), glucose (β = 0.25, SE = 0.12), triglycerides (β = 0.18, SE = 0.05), and high density lipoprotein (HDL-c) (β = −0.10, SE = 0.04); high TVV/moderate PA with BMI (β = 1.98, SE = 0.45), WC (β = 5.43, SE = 1.12), %BF (β = 5.15, SE = 0.72), triglycerides (β = 0.08, SE = 0.04), total cholesterol (β = 0.21, SE = 0.10), low density protein (LDL-c) (β = 0.19, SE = 0.08), and HDL-c (β = −0.07, SE = 0.03); and moderate TVV/insufficient PA with WC (β = 2.68, SE = 1.25), %BF (β = 3.80, SE = 0.81), LDL-c (β = 0.18, SE = 0.09), and HDL-c (β = −0.07, SE = 0.03). Independent of PA levels, a higher TVV was associated with higher amounts of adipose tissue. Higher blood glucose and triglycerides were present in subjects with high TVV and insufficient PA, but not in those with high PA alone. These results affirm the independent cardiometabolic risk of sedentary routines even in subjects with high-levels of PA.

## 1. Introduction

Sedentary behavior (SB) is defined as any waking behavior characterized by an energy expenditure ≤1.5 metabolic equivalents (METs) while in a sitting, reclining, or lying posture [1], and represents a risk factor for cardiometabolic disease [2]. Television viewing (TVV), video game playing, computer use, reading, and sitting while commuting (by bus, train, car, etc.) are examples of SB [1,3]. TVV has been one of the main proxy measures of SB in the domestic setting [2] and is correlated with older age, lower education, an unhealthier lifestyle, type 2 diabetes, osteomuscular disease, cardiovascular disease, and mortality [4,5]. One proposed mechanism is the association of TVV with snack food consumption, which ignores internal satiety signals and generate excessive energy intake [6], and is associated with a loss of localized muscular contraction with sitting [7]. These events can lead to high elevations in glucose, free fatty acids, and triglycerides [7].

Each type of SB shows a distinct sociodemographic, lifestyle, and health profile [7] and this particularity is not always considered. For instance, there are countries such as in Czechia, situated in Central and Eastern Europe, where concepts of health promotion and primary/primordial prevention emerged after the fall of the communist regime in 1989 and improvements are still required [8]. The time spent sitting per day had increased from 58% in 2005 to 62% in 2017 [9] and TVV remains the main passive leisure activity for families [10]. More than 80% watch television on a traditional TV set every day or almost every day and the use of TV still outweighs the use of computers [11], which makes TVV a very representative SB.

The concern to study the prevalence of physical activity (PA) and SB in Czechia is relatively recent [7]. Traditionally considered an “active country” [12], the prevalence of PA in Czech individuals has decreased in recent years [7,8]. In comparison to PA, SB has still received limited attention globally [1], and evidences suggest not only the importance of TVV, but also an interactive relationship between TVV and PA from a cardiometabolic disease prevention perspective [13]. In Czechia, in contrast to the decreasing burden of cardiometabolic risk factors such as hypertension and dyslipidemia, the prevalence of dysglycemia and adiposity have increased [8]. Gaining more information about the relationship between TVV, PA, and cardiometabolic health is especially important to prioritize prevention strategies. The aim of the present study is to evaluate the association between different levels of TVV and the concurrent PA levels with cardiometabolic risk factors in a population based-study of the Czech population.

## 2. Materials and Methods

### 2.1. Study Design and Population

The Kardiovize is a cross-sectional population-based study with a random sample of 1% of adult (25–64 years old) population of Brno, Czech Republic. Brno is the second-largest city in the Czech Republic, with 373,327 residents in 2013. The eligibility criteria included permanent residence in Brno and registration with any of the five state-run health insurance companies operating in the country, covering 91.1% of the population. 

Survey sampling was done in January 2013 with technical assistance using registries from the largest (state-run) health insurance company. A random stratified sample by age and gender of 3300 persons was adjusted for a response rate of 64.4% (as projected from the Czech post-MONICA study). Health insurance companies mailed invitation letters with a description of the study ensuring confidentiality. As the sample size was not reached, a second random sample was done following the same methodology as the first sample. For the second invitation, 3077 invitations were mailed. Based on the two samplings with a total of 6377 randomly selected invitees, the overall response rate was 33.9% [14].

No information on non-respondents was available. A total of 2160 individuals signed their informed consent to participate and were enrolled. For this analysis, subjects with type 1 diabetes were excluded.

The study protocol complied with the Helsinki declaration and all participants signed their informed consent. The Kardiovize study was approved by the ethics committee of St. Anne’s University Hospital, Brno, Czech Republic.

### 2.2. Data Collection

The questionnaire included demographics (e.g., age, education, and marital status), socioeconomic status, cardiovascular risk behaviors (e.g., smoking, nutrition, alcohol consumption, and physical activity), family and personal history, medications, and hospitalizations. Laboratory analyses were performed on 12-h fasting whole blood samples using a Modular SWA P800 analyzer (Roche, Basel, Switzerland). Total cholesterol, triglycerides, and glucose were analyzed by the enzymatic colorimetric method (Roche Diagnostics GmbH, Mannheim, Germany). High-density lipoprotein cholesterol (HDL-c) was analyzed with a homogeneous method for direct measurement without precipitation (Sekisui Medical, Hachimantai, Japan). The low-density lipo-protein cholesterol (LDL-c) level was calculated according to the Friedewald equation when triglyceride levels were below 4.5 mmol/L; if it was higher, LDL-c was analyzed using the homogeneous method for direct measurement (Sekisui Medical, Hachiman-tai, Japan). Blood pressure was measured with the patient alone using an automated office measurement device (BpTRU, model BPM 200; Bp TRU Medical Devices Ltd., Coquitlam, BC, Canada). 

Anthropometric assessment included height, weight, body mass index (BMI) (kg/m^2^), and waist circumference (WC). Height was measured using a stadiometer (SECA 799; SECA, GmbH and Co. KG, Hamburg, Germany) and a manual tape was used to measure waist, hip, and neck circumference. Weight and body composition analyses were performed on a scale with multi-frequency (5, 50, and 250 kHz) bioelectrical impedance analysis using an eight-point tactile electrode system (InBody 370; BIOSPACE Co., Ltd., Seoul, Korea). The direct segmental multi-frequency technique employs the assumption that the human body is composed of five interconnecting cylinders and takes direct impedance measurements from the various body compartments [15]. The spectrum of electrical frequencies predicts the phase angle [16], intracellular water (ICW), and extracellular water (ECW) compartments of the total body water (TBW) in the various body segments. Low-level frequencies rely on the conductive properties of extracellular fluid, whereas at high-level frequencies the conductive properties of both ICW and ECW are instrumental. Lean body mass is estimated as (ICW + ECW)/0.73 [15]. Fat mass is calculated as the difference between the total body weight and lean body mass. The device has an auto-calibration function that is completed each time it is turned on. The In-Body analyzers seem reliable and appear to be a reasonable surrogate in the absence of dual-energy X-ray absorptiometry (DXA) [17]. The model used in the present research in particular (InBody 370) can be found in different studies [18,19,20].

### 2.3. Variables Definition

Self-reported PA was assessed using the long version of the International Questionnaire of Physical Activity (IPAQ). Subjects classified as “high PA” were those who participated in vigorous-intensity activity at least 3 days per week, achieving a minimum of 1500 MET-minutes/week, or 7 or more days per week of any combination of walking, moderate-intensity, or vigorous-intensity activities achieving a minimum of 3000 MET-minutes/week. Subjects classified as “moderate PA” were those who participated in at least 20 min of vigorous physical activity 3 or more days per week, or at least 30 min of moderate-intensity physical activity or brisk walking 5 or more days per week, or 5 or more days per week of any combination of walking, moderate-intensity, or vigorous-intensity activities, achieving a minimum of 600 MET-min/week. Subjects classified as “insufficient PA” were those who did not reach the activity levels listed above. Information about daily TVV time was obtained through the question “How many hours do you spend watching television?” Weekly television watching was estimated by taking a weighted average of daily weekday and weekend activity (weekly television time = (average daily weekday television time × 5) + (average daily weekend television time × 2)). The continuous value was classified as “low” (<2 h/day), “moderate” (between ≥2 to <4 h), and “high” (≥4 h/day) [21,22]. Marital status was classified as “living alone” (including single, divorced, and widowed) or “living as a couple” (including married and other partnerships). Educational level was classified as primary, secondary, and higher (representing the highest level achieved). Household income was expressed in Euros per month and classified as “low” (<1200), “middle” (1200–1800), or “high” (>1800). Smoking status was classified as “non-smoker” or “current smoker” (smoking in any amount during the past year). Alcohol consumption was assessed by the reported alcohol intake of the last week, expressed in a number of standard drinks. One standard drink contained approximately 10 g of ethanol (100–125 mL of wine, 250 mL of beer, or 30 mL of spirits). Participants were classified as a “non-alcohol user” (including abstainers and those who did not drink in the previous 12 months) and “alcohol user” [14].

### 2.4. Data Analysis

All statistical analyses were performed using the STATA software (version 14.0, StataCorp, College Station, TX, USA). The Kolmogorov–Smirnov test was used to assess the normal distribution of variables. Continuous variables with skewed distributions were reported as the median and interquartile range (IQR) and compared using the Mann–Whitney U test. Categorical variable frequency was reported as a percentage and confidence interval (CI) and compared using the chi-squared or Fisher test. To under-stand the impact of TVV on cardiometabolic risk factors across different levels of PA, the participants were classified according to nine exposure groups based on TVV and PA levels: Low TVV/insufficient PA, low TVV/moderate PA, and low TVV/high PA; moderate TVV/insufficient PA, moderate TVV/moderate PA, and moderate TVV/high PA; and high TVV/insufficient PA, high TVV/moderate PA, and high TVV/high PA. Those with low time watching TVV and a high level of PA were considered as the reference group, representing ideal behavior. 

Outcomes were cardiometabolic risk factors, which are expressed as continuous variables, such as BMI, WC, body fat percentage, blood pressure, glucose, triglycerides, total cholesterol, LDL-c, and HDL-c. To assess the independent association of TVV/PA classes with cardiometabolic risk factors, multiple linear regression analyses were used to estimate β coefficients with accompanying standard estimates (SE). Two regression models were presented: Model 1—adjusted for sex and age; and Model 2 for age, sex, educational level, income, alcohol consumption, smoking, and specific medication use. Statistical significance was set at *p* < 0.05.

## 3. Results

### 3.1. Subject’s Characteristics

In total, 2155 subjects were included, 45.2% were men with a median age of 48.0 (IQR 19.0) years. When the subjects were categorized by their PA level, the prevalence of insufficient, moderate, and high PA was 11.8%, 42.1%, and 46.1%, respectively (Table 1). Subjects with an insufficient PA had a higher BMI, WC, body fat percentage, triglycerides, total cholesterol, LDL-c, and low HDL-c than those with higher levels of PA. An insufficient PA was also associated with a higher prevalence of alcohol use, smoking, and use of diuretics. Subjects with a lower education level had a lower prevalence of insufficient PA than those with higher education (Table 1).

The prevalence of low, moderate, and high TVV were 42.6%, 39.7%, and 17.7%, respectively (Table 2). Subjects with a high TVV were older and had a higher BMI, WC, body fat percentage, blood pressure, blood glucose, triglycerides, total cholesterol, LDL-c, and lower HDL-c than those with a lower TVV. High TVV was also associated with a higher prevalence of smoking, insufficient PA, and use of antihypertensive and hypolipidemic medications. Subjects with lower education and income showed a higher TVV than those with higher education and income (Table 2).

### 3.2. Association of Television Viewing/Physical Activity and Cardiometabolic Risk Factors

Compared with subjects with a low TVV and high PA level, those with high TVV were consistently associated with higher amounts and an inappropriate distribution of adipose tissue, independent of their PA level (Table 3). Independent of multiple covariates (Model 2), subjects with a high TVV and low PA level had, on average, a 2.61-higher BMI, a 7.52-cm higher WC, and 6.24%-higher body fat percentage than those with a low TVV and high PA level. Even those with a high TVV and high PA level had a 1.03-higher BMI, 2.42-cm higher WC, and 2.40%-higher body fat percentage than the reference group (Table 3). 

Higher triglycerides were present among subjects with a high TVV in combination with moderate and insufficient PA, but not in those with high PA levels. In addition, higher levels of blood glucose were present in subjects with a high TVV in combination with insufficient PA, but not in those with moderate and high PA levels (Table 3). Other biomarkers, such as total cholesterol, LDL-c, and HDL-c were inconsistently associated with different categories of TVV and PA. Blood pressure did not show any association after multiple adjustments.

## 4. Discussion

In this central European population of adults between 25 to 65 years old with a low prevalence of insufficient PA (11.7%), high TVV was observed in 17.7% of the subjects, and was similar between genders, and showed an increase with age (Table 2). High TVV was strongly and consistently associated with a higher amount and inappropriate distribution of adipose tissue, independent of the level of PA, and with higher values of blood glucose and triglycerides in subjects with insufficient PA. 

The analysis combining TVV/PA levels was also implemented by Eisenmann et al. [23] in high school students from 14 to 18 years, and high TVV levels were identified as a risk factor for adiposity drivers even in combination with moderate and high levels of physical activity. In the present study, despite the achievement of sufficient PA, high TVV was associated with the presence of cardiometabolic risk factors, especially in increasing adiposity levels and insulin-resistance related biomarkers (blood glucose and triglycerides). Consistent with previous studies, Dunstan et al. [22] showed in a large population-based study that TVV was positively associated with glucose, independent of PA level. Lemes et al. [24] showed that the combination of high TVV, characterized by the responses “often and very often”, and PA increased the odds of having metabolic syndrome (MetS) by 89% even after adjustments for confounders. The combination of insufficient PA (<150 min/week) and higher TVV (>2 h/week) was also associated with an increased prevalence of MetS [25]. Suminski et al. [13] found in a similar age sample of 454 adults, an association between continuous TVV hours/week and body fat percentage, but only in the group with insufficient PA.

To the best of our knowledge, this is the first report evaluating the association of different levels of TVV, PA, and body fat percentage by bioimpedance in an adult sample. This study shows that despite achieving a recommended level of PA, there was still a significant association of high TVV time with adiposity drivers, including a higher amount of adipose tissue estimated by body fat percentage and BMI, and inappropriate distribution by WC. Body fat percent was better correlated with diverse categories of TVV and PA than BMI, highlighting how using only BMI as a surrogate of the amount of fat can underperform as a predictor of health [26]. The results also showed an association between moderate and high levels of TVV with the WC being independent of the PA level. Inappropriate distribution of adipose tissue detected by a high WC is well associated with insulin resistance states and inflammation [7]. Insulin resistance is at the intersection of abnormal adiposity and dysglycemia [27]. In this study, the combination of high TVV/insufficient PA was associated with higher values of blood glucose, triglycerides, and lower HDL-c, all characteristics of the insulin resistance states and MetS [28].

Leisure time SB related to TVV has a different relationship with cardiometabolic risk factors than observed with occupational SB (work context) [1]. Unlike occupational SB, TVV typically occurs in the evenings and usually after dinner. It is important to note that prolonged postprandial sedentary time may be particularly detrimental for glucose and lipid metabolism [29]. In addition, habitual TVV is generally characterized by prolonged sitting without breaks in sitting time [30], snacking [31], increased consumption of sugar-sweetened beverages, tobacco smoking (especially initiation) [32,33], and cardiometabolic risk. Unlike leisure time SB, occupational SB is often associated with favorable socio-demographic and lifestyle factors that could mitigate the effect of prolonged sitting time [34].

The influence of SB results from complex interactions among diverse domains, including intrapersonal, interpersonal, and environmental [35]. Nowadays, social restrictions during the COVID-19 pandemic could promote indoor unhealthy behaviors. The U.K. Office of Communications’ latest annual study of media use habits reported that the total video viewing minutes per person per day increased from 4 h and 52 min in 2019, to 6 h and 25 min in April 2020; this included total video including TVV, YouTube devices, Games consoles, and other modalities [36]. In this report, the TVV minutes increased from 154 to 178 for all ages. Specific recommendations should be provided to promote healthy lifestyle routines and to overcome the challenges imposed by the pandemic to improve PA. Examples include engaging in home aerobic and strength training, ordering home exercise equipment, participating in judicious outdoor walking/running, designing realistic daily exercise programs, and establishing new home routines for a healthy lifestyle [37].

Important strengths of the present study include the representativeness of the sample. The city of Brno, a single urban setting, represents the urban population in the Czech Republic [38]. Except for Prague, the capital, with a population over one million, other 10 major cities have populations of approximately 100,000 to 300,000 [39]. Reliable and valid self-administered questionnaires were used to obtain information on PA and TVV and objective data were collected for all cardiometabolic risk factors. Limitations of this study are related to the cross-sectional design, which does not allow a causal relationship to be established. Differences in objective measurements and self-reported behavior might bias results. Future reports should be implemented, especially for objective measurements of TVV.

Most of the studies regarding the association of TVV, PA, and cardiometabolic outcomes in adults are from Australia, United States, Canada, Germany, the United Kingdom, Japan, and Brazil [24]. To the best of our knowledge, this is the first study that focuses on the Czech Republic. The current findings support the importance of TVV and SB in a sample with a majority of physically active subjects (82.3%), which is much higher than comparable studies (from 25% to 58.4%) [13,24,25], and in a nation already considered as an “active country” [32], a country of walkers and cyclists [40]. Unfortunately, the Czech Republic has also experienced a rising prevalence of SB and non-communicable diseases [7], creating an imperative to delineate ways to improve the overall health of the population.

## 5. Conclusions

Higher TVV was associated with the presence of higher adiposity levels independent of PA levels. Higher levels of blood glucose and triglycerides were also presented in subjects with high-time TVV and insufficient PA. Lower HDL-C levels were observed only in the low PA level group. These results highlight the risk of sedentary routines even among subjects with a high level of PA. Current studies and clinical evaluations should consider TVV time, especially now that streaming options are easily available and social isolation due to the COVID-19 pandemic has led to more time spent in the domestic setting.

## Figures and Tables

**Table 1 jcm-11-00545-t001:** Population characteristics according to physical activity level.

	Insufficient	Moderate	High	*p*
*n* (%)	254 (11.8%)	907 (42.1%)	994 (46.1%)	
Sex (% Men)	48.4	42.8	46.7	0.128
Age (years)	50 (19)	48 (19)	48 (20)	0.376
BMI (kg/m^2^)	26.0 (5.0)	25.0 (7.0)	25.0 (6.2)	0.023
WC (cm)	91.0 (22.0)	88.0 (22.0)	89.0 (20.0)	0.006
Body fat percentage (%)	28.0 (13.0)	26.0 (14.0)	24.0 (14.5)	<0.001
Systolic blood pressure (mmHg)	120.3 (22.8)	118.4 (20.6)	118.4 (18.2)	0.114
Diastolic blood pressure (mmHg)	79.4 (13.0)	80.0 (13.0)	79.2 (12.0)	0.217
Glucose (mmol/L)	4.9 (0.7)	4.9 (0.7)	4.9 (0.7)	0.764
Triglycerides (mmol/L)	1.1 (0.9)	1.0 (0.7)	1.0 (0.7)	0.006
Total Cholesterol (mmol/L)	5.2 (1.4)	5.1 (1.3)	5.0 (1.2)	0.009
LDL-c (mmol/L)	3.2 (1.2)	3.0 (1.2)	2.9 (1.2)	0.002
HDL-c (mmol/L)	1.3 (0.5)	1.5 (0.5)	1.4 (0.4)	<0.001
Educational Level (%)				
Primary	18.1 (13.3–22.8)	15.2 (12.8–17.5)	24.5 (21.8–27.1)	<0.001
Secondary	35.8 (29.9–41.7)	36.9 (33.7–40.0)	40.9 (37.8–43.9)	
Higher	46.1 (39.9–52.2)	47.8 (44.5–51.0)	34.6 (31.6–37.5)	
Household income (Euro) (%)				
Low (<1200)	42.9 (36.8–48.9)	40.1 (36.9–43.2)	45.5 (42.4–48.6)	0.006
Middle (1200–1800)	29.4 (23.8–35.0)	30.9 (27.8–33.9)	33.0 (30.0–35.9)	
High (>1800)	27.7 (22.2–33.2)	30 (27.0–32.9)	25.5 (22.7–28.2)	
Living as a couple (%)	60.1 (54.0–66.1)	64.8 (61.6–67.9)	64.2 (61.2–67.1)	0.085
Current smoker (%)	29.9 (24.2–35.5)	20.4 (17.7–23.0)	24.4 (21.7–27.0)	0.005
Alcohol user (%)	87.0 (82.8–91.1)	85.7 (83.4–87.9)	82.1 (79.7–84.4)	0.042
Medications (%)				
Diuretic	11.0 (7.1–14.8)	6.1 (4.5–7.6)	7.7 (6.0–9.3)	0.025
Vasodilator	26.0 (20.6–31.3)	21.2 (18.5–23.8)	23.2 (20.5–25.8)	0.23
Hypoglycemic	6.3 (3.3–9.2)	3.9 (2.6–5.1)	3.3 (2.1–4.4)	0.088
Hypolipidemic	9.4 (5.8–12.9)	9.0 (7.1–10.8)	11.6 (9.6–13.5)	0.161

Continuous variables are represented by median and interquartile range (IQR). Mann–Whitney U test was used to determine different medians. Proportions are present as percent and 95% confidence intervals. Chi-square test was used to determine different proportions. Abbreviations: BMI—body mass index; WC—waist circumference; LDL-c—low density lipoprotein cholesterol; HDL-c—high density lipoprotein cholesterol.

**Table 2 jcm-11-00545-t002:** Population characteristics according to television viewing habits.

	Low	Moderate	High	*p*
*n* (%)	918 (42.6%)	855 (39.7)	382 (17.7)	
Sex (% Men)	44.1	44.7	49.7	0.158
Age (years)	44.0 (19)	49.0 (20)	52.0 (17)	<0.001
BMI (kg/m^2^)	25.0 (6.0)	26.0 (6.0)	27.0 (7.0)	<0.001
Waist circumference (cm)	86.0 (19.0)	90.0 (20.0)	95.0 (20.0)	<0.001
Body fat percentage (%)	24.0 (12.0)	26.0 (14.0)	29.0 (15.0)	<0.001
Systolic blood pressure (mmHg)	115.8 (18.4)	119.4 (16.6)	123.5 (20.2)	<0.001
Diastolic blood pressure (mmHg)	78.6 (12.4)	80.0 (12.8)	80.7 (12.0)	<0.001
Glucose (mmol/L)	4.8 (0.7)	5.0 (0.7)	5.0 (0.8)	<0.001
Triglycerides (mmol/L)	0.9 (0.7)	1.0 (0.7)	1.3 (0.9)	<0.001
Total cholesterol (mmol/L)	5.0 (1.2)	5.1 (1.3)	5.2 (1.5)	<0.001
LDL-c (mmol/L)	2.9 (1.1)	3.0 (1.2)	3.2 (1.3)	<0.001
HDL-c (mmol/L)	1.5 (0.5)	1.4 (0.5)	1.4 (0.4)	<0.001
Educational Level (%)				
Primary	13.5 (11.2–15.7)	20.7 (17.9–23.4)	33.0 (28.2–37.7)	<0.001
Secondary	37.4 (34.2–40.5)	37.7 (34.4–40.9)	43.5 (38.5–48.4)	
Higher	49.0 (45.7–52.2)	41.6 (38.3–44.9)	23.6 (19.3–27.8)	
Household income (Euro) (%)				
Low (<1200)	38.1 (34.9–41.2)	42.5 (39.1–45.8)	55.4 (50.4–60.3)	<0.001
Middle (1200–1800)	32.4 (29.3–35.4)	32.9 (29.7–36.0)	27.3 (22.8–31.7)	
High (>1800)	20.5 (17.8–23.1)	24.6 (21.7–27.4)	17.3 (13.5–21.0)	
Living as a couple (%)	61.7 (58.5–64.8)	64.9 (61.7–68.1)	59.4 (54.4–64.3)	0.139
Physical activity level (%)				
Insufficient	8.9 (7.0–10.7)	12.6 (10.3–14.8)	16.7 (12.9–20.4)	<0.001
Moderate	42.4 (39.2–45.6)	43.3 (39.9–46.6)	38.7 (33.8–43.5)	
High	48.7 (45.4–51.9)	44.1 (40.7–47.4)	44.5 (39.5–49.4)	
Smokers (%)	19.5 (16.9–22.0)	22.2 (19.4–24.9)	35.6 (30.8–40.4)	<0.001
Alcohol user (%)	84.2 (81.8–86.5)	84.7 (82.2–87.1)	82.7 (78.9–86.4)	0.683
Medications (%)				
Diuretic	4.8 (3.4–6.1)	8.1 (6.2–9.9)	11.3 (8.1–14.4)	<0.001
Vasodilator	16.0 (13.6–18.3)	24.2 (21.3–27.0)	34.5 (29.7–39.2)	<0.001
Hypoglycemic	2.8 (1.7–3.8)	4.6 (3.2–6.0)	5.0 (2.8–7.1)	0.083
Hypolipidemic	7.7 (5.9–9.4)	11.5 (9.3–13.6)	12.8 (9.4–16.1)	0.005

Continuous variables are represented by median and interquartile range (IQR). Mann–Whitney U test was used to determine different medians. Proportions are present as percent and 95% confidence intervals. Chi-square test was used to determine different proportions. Abbreviations: BMI—body mass index; WC—waist circumference; LDL-c—low density lipoprotein cholesterol; HDL-c—high density lipoprotein cholesterol.

**Table 3 jcm-11-00545-t003:** Association between Televidion Viewing (TVV), physical activity level (PA), and cardiometabolic factors (*n* = 2155).

Cardiometabolic Factors	Classification	Model 1 β (SE)	Model 2 β (SE)
BMI (kg/m^2^)	Low TVV/Insufficient PA	−0.04 (0.57)	0.30 (0.57)
	Low TVV/Moderate PA	−0.17 (0.33)	−0.24 (0.56)
	Moderate TVV/Insufficient PA	0.83 (0.51)	0.93 (0.50)
	Moderate TVV/Moderate PA	0.27 (0.33)	0.40 (0.33)
	Moderate TVV/High PA	0.73 (0.33) ^a^	0.53 (0.33)
	High TVV/Insufficient PA	2.93 (0.64) ^b^	2.61 (0.63) ^b^
	High TVV/Moderate PA	2.10 (0.45) ^b^	1.98 (0.45) ^b^
	High TVV/High PA	1.49 (0.43) ^b^	1.03 (0.43) ^a^
WC (cm)	Low TVV/Insufficient PA	0.84 (1.42)	1.65 (1.41)
	Low TVV/Moderate PA	−0.56 (0.82)	−0.10 (0.82)
	Moderate TVV/Insufficient PA	2.42 (1.27)	2.68 (1.26) ^a^
	Moderate TVV/Moderate PA	1.35 (0.84)	1.67 (0.83) ^a^
	Moderate TVV/High PA	0.77(1.44)	−0.47 (1.44)
	High TVV/Insufficient PA	8.32 (1.60) ^b^	7.52 (1.58) ^b^
	High TVV/Moderate PA	5.74 (1.12) ^b^	5.43 (1.12) ^b^
	High TVV/High PA	3.58 (1.07) ^b^	2.42 (1.07) ^a^
Body fat percentage (%)	Low TVV/Insufficient PA	2.04 (0.92) ^a^	2.69 (0.91) ^b^
	Low TVV/Moderate PA	1.07 (0.53) ^a^	1.44 (0.52) ^b^
	Moderate TVV/Insufficient PA	3.64 (0.82) ^b^	3.80 (0.81) ^b^
	Moderate TVV/Moderate PA	1.71 (0.54) ^b^	1.95 (0.53) ^b^
	Moderate TVV/High PA	1.41 (0.54) ^b^	1.08 (0.54) ^a^
	High TVV/Insufficient PA	6.82 (1.04) ^b^	6.24 (1.02) ^b^
	High TVV/Moderate PA	5.37 (0.73) ^b^	5.15 (0.72) ^b^
	High TVV/High PA	3.29 (0.69) ^b^	2.40 (0.69) ^b^
Systolic blood pressure (mmHg)	Low TVV/Insufficient PA	−0.98 (1.69)	−0.64 (1.62)
	Low TVV/Moderate PA	−1.34 (0.98)	−0.93 (0.94)
	Moderate TVV/Insufficient PA	1.19 (1.51)	0.61 (1.41)
	Moderate TVV/Moderate PA	−0.05 (0.99)	−0.29 (0.95)
	Moderate TVV/High PA	−0.01 (0.99)	−0.87 (0.95)
	High TVV/Insufficient PA	4.47 (1.88) ^a^	2.26 (1.83)
	High TVV/Moderate PA	2.82 (1.34) ^a^	0.85 (1.29)
	High TVV/High PA	0.27 (1.27)	−1.41 (1.23)
Diastolic blood pressure (mmHg)	Low TVV/Insufficient PA	−0.16 (1.06)	−0.20 (1.03)
	Low TVV/Moderate PA	−0.13 (0.62)	−0.02 (0.16)
	Moderate TVV/Insufficient PA	1.18 (0.95)	0.72 (0.92)
	Moderate TVV/Moderate PA	0.85 (0.62)	0.57 (0.61)
	Moderate TVV/High PA	−0.24 (0.62)	−0.73 (0.61)
	High TVV/Insufficient PA	1.78 (1.19)	0.72 (1.17)
	High TVV/Moderate PA	1.86 (0.84) ^a^	0.82 (0.82)
	High TVV/High PA	−0.25 (0.80)	−1.07 (0.79)
Glucose (mmol/L)	Low TVV/Insufficient PA	−0.16 (0.11)	−0.13 (0.11)
	Low TVV/Moderate PA	−0.04 (0.06)	−0.01 (0.06)
	Moderate TVV/Insufficient PA	0.06 (0.10)	0.03 (0.09)
	Moderate TVV/Moderate PA	0.04 (0.06)	0.04 (0.06)
	Moderate TVV/High PA	0.09 (0.06)	0.04 (0.06)
	High TVV/Insufficient PA	0.37 (0.12) ^b^	0.25 (0.12) ^a^
	High TVV/Moderate PA	0.20 (0.09) ^a^	0.12 (0.08)
	High TVV/High PA	0.13 (0.08)	0.04 (0.08)
Triglycerides (mmol/L)	Low TVV/Insufficient PA	0.005 (0.05)	0.01 (0.05)
	Low TVV/Moderate PA	0.02 (0.03)	0.03 (0.03)
	Moderate TVV/Insufficient PA	0.05 (0.04)	0.04 (0.04)
	Moderate TVV/Moderate PA	0.04 (0.03)	0.04 (0.03)
	Moderate TVV/High PA	0.03 (0.03)	0.009 (0.03)
	High TVV/Insufficient PA	0.23 (0.05) ^b^	0.18 (0.05) ^b^
	High TVV/Moderate PA	0.11 (0.04) ^b^	0.08 (0.04) ^a^
	High TVV/High PA	0.08 (0.04) ^a^	0.04 (0.03)
Total cholesterol (mmol/L)	Low TVV/Insufficient PA	0.07 (0.13)	0.06 (0.12)
	Low TVV/Moderate PA	0.11 (0.07)	0.13 (0.07)
	Moderate TVV/Insufficient PA	0.19 (0.11)	0.14 (0.11)
	Moderate TVV/Moderate PA	0.21 (0.01) ^b^	0.18 (0.07) ^a^
	Moderate TVV/High PA	0.01 (0.07)	0.02 (0.07)
	High TVV/Insufficient PA	0.16 (0.14)	0.17 (0.14)
	High TVV/Moderate PA	0.25 (0.10) ^a^	0.21 (0.10) ^a^
	High TVV/High PA	0.04 (0.09)	0.03 (0.09)
LDL-c (mmol/L)	Low TVV/Insufficient PA	0.16 (0.10)	0.14 (0.10)
	Low TVV/Moderate PA	0.10 (0.06)	0.12 (0.06) ^a^
	Moderate TVV/Insufficient PA	0.22 (0.09) ^a^	0.18 (0.09) ^a^
	Moderate TVV/Moderate PA	0.13 (0.06) ^a^	0.11 (0.06)
	Moderate TVV/High PA	0.005 (0.06)	0.02 (0.06)
	High TVV/Insufficient PA	0.19 (0.12)	0.19(0.11)
	High TVV/Moderate PA	0.22 (0.08) ^b^	0.19(0.08) ^a^
	High TVV/High PA	0.05 (0.08)	0.05 (0.07)
HDL-c (mmol/L)	Low TVV/Insufficient PA	−0.09 (0.04) ^a^	−0.09 (0.04) ^a^
	Low TVV/Moderate PA	−0.01 (0.02)	−0.01 (0.02)
	Moderate TVV/Insufficient PA	−0.07 (0.03) ^a^	−0.07 (0.03) ^a^
	Moderate TVV/Moderate PA	−0.03 (0.02)	−0.03 (0.02)
	Moderate TVV/High PA	−0.004 (0.02)	−0.006 (0.02)
	High TVV/Insufficient PA	−0.10 (0.04) ^a^	−0.10 (0.04) ^a^
	High TVV/Moderate PA	−0.04 (0.03)	−0.03 (0.03)
	High TVV/High PA	−0.04 (0.03)	−0.04 (0.03)

^a^ *p* < 0.05; ^b^ *p* < 0.01. Continuous variables are represented by median and IQR. Mann–Whitney U test was used to determine different medians. Proportions are present as percent and 95% confidence intervals. Chi-square test was used to determine different proportions. Abbreviations: BMI—body mass index; WC—waist circumference; LDL-c—low density lipoprotein cholesterol; HDL-c—high density lipoprotein cholesterol.

## Data Availability

The data presented in this study are available upon request from the corresponding author. The data are not publicly available.

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
