# Peer review of "The Combined Effects of Television Viewing and Physical Activity on Cardiometabolic Risk Factors: The Kardiovize Study"

_jcm, 2022, doi:10.3390/jcm11030545_

Round 1

Reviewer 1 Report

The authors investigated the association between television viewing/physical activity interactions and cardiometabolic risk in an adult European population. Results showed the independent cardiometabolic risk of sedentary routines even in subjects with high levels of PA.

The manuscript deals with a current and very interesting topic.

I have only a few comments to make.

-The various sections of the manuscript are well written; however, the introduction is too short and should be completed to fully clarify the current level of knowledge. 500 to 600 words (no less than 400 and no more than 700) are enough to write an excellent introduction. Your introduction is less than 300 words.

-Paragraphs 2.1 and 2.2 have the same title. Correct.

-Briefly describe the technical characteristics of InBody, Korea device (in Data collection). What is the calibration method to ensure validity (accuracy and precision) of the bioimpedance measurements? What is the technical error of measurement in vivo? Provide readers with a concise description of what this BIA device measures. In particular, what are the measurements detected by this tool? Do they directly measure the raw bioimpedance parameters (e.g., R, Xc and phase angle)? Again, what equation was used to estimate body composition parameters? Is it an equation developed using the Inbody device or an instrument that works with similar characteristics (frequency and technologies)? Please go deeper into the topic with this reference: https://www.mdpi.com/1105902

Author Response

We thank the reviewer for the valuable suggestions. They greatly improved the material.

Reviewer 2 Report

The study is correctly designed and the results are clearly and transparently presented.The set goals correspond to the conclusions.The findings are interesting and innovative, and underline the important aspect of the negative impact of the Covid- 19 pandemic on increasing cardiovascular risk.

I have no comments for work.

Author Response

Thanks to the reviewer. A new version has been submitted.

Reviewer 3 Report

I cannot see the novelty of this study. This topic has been intensively studied previously... Authors should present stronger significance of study and novelty.

Author Response

We thank the reviewer for the valuable suggestions. They greatly improved the material.

We believe that, especially with the new introduction, we can better contextualize the relevance of the study in Czechia, as well as study gaps.

Round 2

Reviewer 1 Report

Thanks for your replies!

Reviewer 3 Report

I suggest that the manuscript should be accepted as it is.